

# What affects power to estimate speciation rate shifts?

Ullasa Kodandaramaiah[*] and Gopal Murali[*]

IISER-TVM Centre for Research and Education in Ecology and Evolution (ICREEE), School of Biology, Indian Institute of Science Education and Research Thiruvananthapuram, Thiruvananthapuram, India
[*] These authors contributed equally to this work.

## ABSTRACT

The development of methods to estimate rates of speciation and extinction from time-calibrated phylogenies has revolutionized evolutionary biology by allowing researchers to correlate diversification rate shifts with causal factors. A growing number of researchers are interested in testing whether the evolution of a trait or a trait variant has influenced speciation rate, and three modelling methods—BiSSE, MEDUSA and BAMM—have been widely used in such studies. We simulated phylogenies with a single speciation rate shift each, and evaluated the power of the three methods to detect these shifts. We varied the degree of increase in speciation rate (speciation rate asymmetry), the number of tips, the tip-ratio bias (ratio of number of tips with each character state) and the relative age in relation to overall tree age when the rate shift occurred. All methods had good power to detect rate shifts when the rate asymmetry was strong and the sizes of the two lineages with the distinct speciation rates were large. Even when lineage size was small, power was good when rate asymmetry was high. In our simulated scenarios, small lineage sizes appear to affect BAMM most strongly. Tip-ratio influenced the accuracy of speciation rate estimation but did not have a strong effect on power to detect rate shifts. Based on our results, we provide suggestions to users of these methods.

## INTRODUCTION

Much as the advent of methods to infer phylogenies (*Edwards & Cavalli-Sforza, 1963*; *Sokal & Sneath, 1963*; *Camin & Sokal, 1965*; *Hennig, 1965*) led to a spectacular revolution in evolutionary biology, the arrival of mathematical methods to estimate divergence times from molecular phylogenies has offered unprecedented novel insights into macroevolution. More recently, a seminal innovation has been the development of tools to estimate rates of speciation and extinction from time-calibrated phylogenies (*Nee, May & Harvey, 1994*). Until such methods became available, our understanding of macroevolutionary patterns and processes of diversification largely relied on the fossil record, which is incomplete for most taxa (*Benton, Wills & Hitchin, 2000*; *Quental & Marshall, 2010*) and virtually non-existent for many soft-bodied life forms (*Donoghue & Purnell, 2009*). However, the possibility of using dated phylogenies of extant taxa to shed light on macroevolutionary history is appealing, and there has been a phenomenal interest in applying these methods to

Corresponding author
Ullasa Kodandaramaiah,
ullasa@iisertvm.ac.in

understand fundamental questions such as how the mode and tempo of diversification have been influenced by causal factors, for instance, 'key innovations' (e.g., *Hunter & Jernvall, 1995*; *Hodges & Arnold, 1995*; *Near et al., 2012*; *Rainford et al., 2014*; *Peña & Espeland, 2015*; *Sahoo et al., 2017*), biogeography (e.g., *Kozak, Weisrock & Larson, 2006*; *Wahlberg et al., 2009*; *Dunn et al., 2010*; *Sanders, Mumpuni & Lee, 2010*; *Sundue Michael, Testo Weston & Ranker Tom, 2015*) and climate change (e.g., *Jansson & Davies, 2008*; *Dunn et al., 2010*; *Arakaki et al., 2011*; *Ezard et al., 2011*; *Near et al., 2012*; *Xiang et al., 2014*).

The diversification rate of a lineage is the difference between its speciation rate $\lambda$ and extinction rate $\mu$. Testing hypothesis of diversification rate variation is underpinned by the ability to decouple and accurately estimate these rates. Estimates of extinction rates from phylogenies of extant taxa appear to be error prone (*Rabosky, 2010*; *Laurent, Robinson-Rechavi & Salamin, 2015*; *May & Moore, 2016*; but see *Stadler, 2013*; *Beaulieu & O'meara, 2016*). On the other hand, speciation rate estimates are generally considered to be more robust. One of the most common themes in macroevolutionary studies over the last decade has been to test whether a trait (or trait variant) has increased speciation rates (e.g., *Sundue Michael, Testo Weston & Ranker Tom, 2015*; *Claramunt et al., 2012*; *Escudero et al., 2012*; *Litsios et al., 2012*; *Horn James et al., 2014*; *Rainford et al., 2014*; *Xiang et al., 2014*; *Gubry-Rangin et al., 2015*; *Igea et al., 2017*; *Wiens, 2017*; *Sahoo et al., 2017*; *Seeholzer, Claramunt & Brumfield, 2017*), with a suite of analytical tools providing the framework to infer speciation rate variation across the phylogeny.

Historically, analyses testing the effect of a trait on diversification relied on comparisons of species richness of sister clades (e.g., *Mitter, Farrell & Wiegmann, 1988*; *Zeh, Zeh & Smith, 1989*). This method cannot distinguish between $\lambda$ and $\mu$, is prone to Type II error (non-detection of significant diversification rate differences), and does not effectively utilize information from clades with mixed character states (*Maddison, Midford & Otto, 2007*). The most recent methods aim to utilize information in the branching patterns of dated phylogenies to decouple $\lambda$ from $\mu$ (*Stadler, 2011*). The BiSSE (Binary State Speciation and Extinction) (*Maddison, Midford & Otto, 2007*) modelling approach has been especially popular for hypothesis testing because it estimates $\lambda$ and $\mu$ associated with character states, i.e., state-dependent diversification rates. BISSE specifies a stochastic model where $\lambda$ and $\mu$ can depend on the character state of a lineage at each time point, and the rates of character state change are allowed to vary (*Maddison, Midford & Otto, 2007*; *FitzJohn, Maddison & Otto, 2009*). Inferences about speciation and extinction rates in relation to character state are made by comparing the maximum likelihood scores of competing models using likelihood ratio tests or AIC (Akaike Information Criterion) scores. BISSE requires a completely resolved dated phylogeny with information on character states of tips as input, and can take into account incomplete sampling (*FitzJohn, Maddison & Otto, 2009*).

While BiSSE only models binary discrete character states (for example presence or absence of a trait; two states of a trait), extensions of BiSSE can handle other types of data. MuSSE (Multiple SSE; *FitzJohn, 2012*) can deal with multiple discrete character states, while QuaSSE (Quantitative SSE; *FitzJohn, 2010*) allows testing the effect of quantitative traits. GeoSSE (Geographic SSE; *Goldberg, Lancaster & Ree, 2011*) tests region-dependent diversification, while BiSSE-ness (BiSSE-node enhanced state shift; *Magnuson-Ford &*

*Otto, 2012*) and Cladogenetic SSE (ClaSSE; *Goldberg & Igić, 2012*) integrate cladogenetic and anagenetic trait evolution. *Beaulieu & O'meara (2016)* proposed the HiSSE (Hidden States SSE) model, which attempts to account for unmeasured ('hidden') factors impacting diversification rates of a known trait or character state.

In contrast to the BiSSE family of methods, character-independent diversification methods attempt to identify the number and location of rate shifts in speciation and extinction across the tree, without *a priori* information on the mechanism of rate variation. Once the locations of rate shifts are found, the researcher can test for associations with traits of interest. MEDUSA (Modeling Evolutionary Diversification Using Stepwise Akaike Information Criterion; *Alfaro et al., 2009*), is one such framework that has been very popular. MEDUSA incrementally assigns rate shifts to all branches of the tree, and uses stepwise AIC to determine the number and location(s) of rate shifts that best fit the data. Rate shift estimates are thus agnostic of the cause of rate variation among lineages.

BAMM (Bayesian Analysis of Macroevolutionary Mixtures; *Rabosky, 2014*) is the most widely used character-independent diversification method. BAMM assumes that $\lambda$ and $\mu$ are heterogeneous across the phylogeny, and that changes in these parameters across branches occur under a compound Poisson process. It uses reversible-jump Markov chain Monte Carlo to explore models varying in the number of shifts in diversification parameters. Estimates of $\lambda$ and $\mu$, and inferences on the number of rate shifts are based on posterior distributions. Both BAMM & MEDUSA require a dated phylogeny and can accommodate incomplete sampling.

Although BiSSE, BAMM and MEDUSA have been very popular, recent critical evaluations of their performance have highlighted potential shortcomings particular to each method. Using empirical datasets, *Rabosky & Goldberg (2015)* found that BiSSE is prone to high Type I error rates, wherein diversification-neutral traits are often found to be significantly associated with speciation rate. Surprisingly, such false associations appear to be detected even for traits with weak phylogenetic signal (*Rabosky & Goldberg, 2015*). The ability of the BiSSE method to detect state-specific diversification rates has been shown to be affected by factors such as tree size (number of tips) (*Davis, Midford & Maddison, 2013*; *Gamisch, 2016*), tree age (*Simpson et al., 2018*) and tip-ratio bias (i.e., ratio of tips with one character state versus another) (*Davis, Midford & Maddison, 2013*): the method appears to perform better with large trees and low tip-ratio biases. *May & Moore (2016)* used extensive simulations to understand the statistical behavior of MEDUSA, and showed that the method is prone to a very high rate of false inferences of rate shifts (ca. 30% on average), and that the estimated diversification parameters are biased. The probability of rate shift detection in MEDUSA depends on the number of terminals in the tree (*Laurent, Robinson-Rechavi & Salamin, 2015*). *Moore et al. (2016)* showed that the accuracy of BAMM is strongly affected by the priors specified, and that the estimates diversification rate parameters are unreliable (although see *Rabosky, Mitchell & Chang, 2017*). *Meyer & Wiens (2018)* found that BAMM underestimated the number of rate shifts, and overestimated diversification rates for small clades.

Analyses of simulated phylogenies have proved to be a valuable tool to assess the performance of the various modelling approaches. We simulated phylogenies with a single

**Figure 1 Flowchart of the simulation process.** Flowchart of the simulation process used to generate a composite tree with a single speciation rate shift. A *basetree* with a target age ($B_{age}$) and speciation rate $\lambda_0$ was simulated using TESS, and pruned at a target *subtree* age ($S_{age}$). A *subtree* with speciation rate $\lambda_1$ and the target $S_{age}$ was simulated independently and grafted on to the *basetree* at the pruned node.

speciation rate shift each, and evaluated the power of BiSSE, BAMM & MEDUSA to detect these shifts. We varied the degree of increase in speciation rate, the number of tips, the tip-ratio bias and the relative age in relation to tree age when the rate shift occurred. We found that all methods have good power to detect rate shifts under many conditions, but also identified some scenarios under which these methods have low power.

# MATERIALS AND METHODS

## Simulation of phylogenetic trees with single diversification rate shifts

### General procedure

The basic workflow of the simulation process is outlined in Fig. 1. To obtain a phylogeny with a single shift in speciation rate, we first simulated two trees—a *basetree* and a *subtree* wherein the *subtree* had a greater speciation rate ($\lambda_1$) compared to that of the *basetree* ($\lambda_0$). We used the package *TESS* (*Höhna, May & Moore, 2015*) to simulate trees in R (*Team, 2016*). The package implements tree simulation based on a global, time-dependent birth-death process conditioned either on number tips or age (*Höhna, May & Moore, 2015*).

We generated phylogenetic trees, both *basetree* and *subtree,* under a constant birth-death process by conditioning on age using the function *tess.sim.age*, which simulates trees given the age ($B_{age}$ or $S_{age}$) and diversification rate parameters ($\lambda$ and $\mu$). A *subtree* of a given age ($S_{age}$) was grafted onto the *basetree* following pruning of a randomly chosen *basetree* clade with approximately the same age ($S_{age} \pm 2.5\%$) (Fig. 1), using a custom written function (available in Figshare). This generated a composite tree with a single speciation rate shift ($\lambda_0$ to $\lambda_1$). The relative age of the *subtree* in relation to that of the overall tree was varied by varying the age of the *subtree*. We ran two simulation sets, each with different goals.

### Simulation Set 1: Effects of speciation rate asymmetry, overall tree size and relative subtree age

Simulation Set 1 aimed to test the relative effects of speciation rate asymmetry ($\lambda_1/\lambda_0$), relative *subtree* age ($100 * S_{age}/B_{age}$) and overall tree size (Table 1), and therefore these three parameters were varied. Tree size variation was incorporated by defining specific target overall tree size classes ($50 \pm 10$, $100 \pm 10$, $150 \pm 10$, $200 \pm 10$, $300 \pm 10$ and $500 \pm 10$) *a priori*, and for each tree size class, simulating trees with all combinations of relative *subtree*

**Table 1  Parameter values of variables in the simulations.** Range of parameter values of variables used for simulation of trees for Simulation Sets 1 and 2.

| Variable | Values in simulation set 1 | Values in simulation set 2 |
|---|---|---|
| Basetree age ($B_{age}$) | 3–35 | 3–35 |
| Subtree age ($S_{age}$) | 20, 40, 60% of *basetree* | 30–70% of *basetree* |
| Basetree speciation rate ($\lambda_0$) | 0.20 | 0.20, 0.26, 0.28 |
| Speciation rate asymmetry ($\lambda_1/\lambda_0$) | 1.5, 2.5, 3.5, 4.5, 5.5 | 2 |
| Basetree extinction rate ($\mu_0$) | 0.05 | 0.05 |
| Subtree extinction rate ($\mu_1$) | 0.05 | 0.05 |
| Overall tree size class | 50 ± 10, 100 ± 10, 150 ± 10, 200 ± 10, 300 ± 10 and 500 ± 10 | 50 ± 10, 100 ± 10, 200 ± 10, 400 ± 10 and 800 ± 10 |

age (20%, 40% and 60%) and speciation rate asymmetries (1.5X, 2.5X, 3.5X, 4.5X and 5.5X). Speciation rate asymmetry was varied by altering $\lambda_1$, while keeping extinction rates constant ($\mu_0=\mu_1$). Thus, higher asymmetry values indicate a greater degree of speciation rate increase in the *subtree*. Pilot runs indicated that a $\lambda_0$ value of 0.2 and $B_{age}$ values between 5 and 35 units generated trees with the required parameters. Details of the pilot runs are described in Supplemental Information 1. We simulated 50 composite replicate trees each for all combinations of tree size class, relative *subtree* age and speciation rate asymmetry (Table 1). Details of how each replicate tree was simulated are described in Supplemental Information 1. Thus, this simulation set comprised 1,500 trees each for three relative *subtree* ages. Simulated trees were subsequently used in the diversification analyses where BAMM, MEDUSA and BiSSE were used to detect the simulated rate shifts (see 'Estimation of diversification rate parameters and power of modelling methods').

### Simulation Set 2: effects of tip-ratio and number of tips in basetree and subtree

Simulation Set 2 aimed to test the relative effects of tip-ratio bias (*basetree:subtree* number of tips), *basetree* size and *subtree* size. We simulated trees with three tip-ratio values (1:9, 9:1 and 1:19), all with a 2X speciation rate asymmetry. We chose 2X because our pilot simulations indicated that power was neither very high nor very low. For each tip-ratio value, we simulated trees with different overall tree sizes (50 ± 10, 100 ± 10, 200 ± 10, 400 ± 10 and 800 ± 10). In order to achieve the target tip numbers and tip-ratios, $B_{age}$ was allowed vary between 5 and 35 units and relative *subtree* age between 30 and 70%, while $\lambda_0$ had values 0.2, 0.26 or 0.28 (Table 1). These ranges were decided based on pilot simulations described in Supplemental Information 1. Hundred replicate trees were generated for each parameter combination (overall tree size class, tip-ratio value), with all replicates for a given tip-ratio plus tree size combination having the same $B_{age}$, $\lambda_0$ and relative *subtree* age (see Supplemental Information 1). We were unable to simulate trees with 1:9 ratio for the 100 ± 10 size class, and therefore used a 1:8 ratio as an approximation.

## Estimation of diversification rate parameters and power of modelling methods

We used BiSSE, MEDUSA and BAMM to estimate diversification rate parameters ($\lambda_0$, $\mu_0$, $\lambda_1$ and $\mu_1$) of the simulated composite trees and detect rate shifts. Power was calculated for each parameter combination as the proportion of the replicate trees in which a significant rate shift was detected at the node where the *subtree* was attached to the *basetree*.

### BiSSE

For the BiSSE analysis, a character was assigned to be present in all *subtree* tips, but absent in all *basetree* tips. Therefore, diversification rate estimates of the BiSSE model will reflect the speciation and extinction rate estimates of the *subtree* (i.e., $\lambda_1$ and $\mu_1$) and *basetree* ($\lambda_0$ and $\mu_0$). We recorded AIC scores and the maximum likelihood values for all possible BiSSE models for a given tree:

I. $\lambda_1 \neq \lambda_0$, $\mu_1 \neq \mu_0$, $q_{01} \neq q_{10}$ (All rates unequal; full model)
II. $\lambda_1 = \lambda_0$, $\mu_1 \neq \mu_0$, $q_{01} \neq q_{10}$ (Only speciation rates equal)
III. $\lambda_1 \neq \lambda_0$, $\mu_1 = \mu_0$, $q_{01} \neq q_{10}$ (Only extinction rates equal)
IV. $\lambda_1 \neq \lambda_0$, $\mu_1 \neq \mu_0$, $q_{01} = q_{10}$ (Only transition rates equal)
V. $\lambda_1 = \lambda_0$, $\mu_1 = \mu_0$, $q_{01} \neq q_{10}$ (Only speciation and extinction rates equal)
VI. $\lambda_1 \neq \lambda_0$, $\mu_1 = \mu_0$, $q_{01} = q_{10}$ (Only extinction and transition rates equal)
VII. $\lambda_1 = \lambda_0$, $\mu_1 \neq \mu_0$, $q_{01} = q_{10}$ (Only speciation and transition rates equal)
VIII. $\lambda_1 = \lambda_0$, $\mu_1 = \mu_0$, $q_{01} = q_{10}$ (All rates equal; null model)

Because we were interested in testing whether speciation rates differ between *basetree* and *subtree,* we chose the best among the models where the speciation rates are unequal (model I, III, IV and VI) based on the minimum AIC score, and compared this with the best model (selected based on minimum AIC score) where speciation rates are equal (model II, V, VII and VIII) using a likelihood ratio test. If $P < 0.05$, the rate shift was considered to have been detected, while the rate shift was not considered detected if $P \geq 0.05$. Our motivation was to mirror how a typical researcher may infer a speciation rate shift using BiSSE on empirical data sets. Using the approach outlined above, a user will infer a shift in speciation rate, disregarding whether or not there are changes in extinction and transition rates. In the Results section, we refer to this approach of model selection as the 'approach 1'.

Because the true model, especially with respect to transition rates, is unclear, we also calculated power by comparing models III with V (referred to as 'approach 2'), as well as VI with VIII (referred to as 'approach 3').

### MEDUSA

We performed MEDUSA analyses using the function *MEDUSA* (available from GitHub as an R package https://github.com/josephwb/turboMEDUSA, downloaded August 2017). We specified the model of tree evolution to be a birth-death process to estimate the diversification rate parameters. We recorded the node where the rate shift was detected and the estimated diversification rates. The diversification rate shift in a tree was considered to be correctly detected if a model with >1 diversification rates was chosen as the best model, and the rate shift was located at the node where the *subtree* was attached to the *basetree*.

### BAMM

We performed BAMM analyses with the default parameter settings using the control file available from the BAMM website (http://bamm-project.org/quickstart.html#control-file accessed: August 2017). The priors used to estimate the speciation and extinction rates were generated using the *setBAMMpriors* for each tree using the *BAMMtools* package (*Rabosky et al., 2014*) in R. We ran the MCMC analysis for 2 million iterations and checked for convergence using ESS metrics (>200). The *bammdata* object was generated using the *getEventData* function from the *BAMMtools* package after discarding the first 10% of samples as burnin. The *bammdata* object was then used to calculate the diversification rates.

We estimated speciation and extinction rates for the *subtree* ($\lambda_1$ and $\mu_1$) as the average rate of the clade using the function *getCladeRates* from the BAMMtools package (*Rabosky et al., 2014*). We estimated $\lambda_0$ and $\mu_0$ using the same function by specifying the common ancestor node of the composite tree, but excluding the rates of the *subtree*. The best diversification rate shift configuration was identified using the function *getBestShiftConfiguration* with the expected number of shifts set to 1. The diversification rate shift was considered to have been correctly detected if the best rate shift configuration included a rate shift at the node where the *subtree* was attached to the *basetree*.

### Accuracy of estimated asymmetry and speciation rates

To compare the difference between the true and estimated asymmetry in Simulation Set 1, we calculated the estimated asymmetry as the median of the $\lambda_1/\lambda_0$ values of the 50 replicate trees for a given parameter combination, and error was calculated as.

Error in estimation of asymmetry = (Estimated asymmetry-True asymmetry)/True asymmetry.

Thus, a value of 0 represents no error, values <0 represent underestimation and those >0 represent overestimation.

We calculated error in estimation of $\lambda_0$ and $\lambda_1$ for trees in Simulation Set 2 as the median of estimated/true $\lambda$ for the 100 replicate trees of each combination of tree size class and tip-ratio. This was done independently for *subtree* and *basetree*. Thus, a value of 1 represents no error, values >1 indicate overestimation of $\lambda$, and values <1 indicate underestimation.

## RESULTS

### Effect of speciation rate asymmetry, overall tree size and relative *subtree* age on power

In analyses of trees from Simulation Set 1, power tended to increase both with asymmetry and tree size across all three *subtree* ages (Fig. 2), but the overall effect of asymmetry and tree size depended on *subtree* age ($S_{age}$).

For BiSSE, when using 'approach 1' model selection, power was generally high (>0.75) for all combinations except when tree size was 50, and 1.5X was in combination with 20% $S_{age}$ (Figs. 1A, 1D 1G). A tendency for increase in power with increasing tree size was most apparent when $S_{age}$ was 60%. Results were similar when power was calculated using two additional approaches of model selection (approaches 2 & 3; Supplemental Information 2).

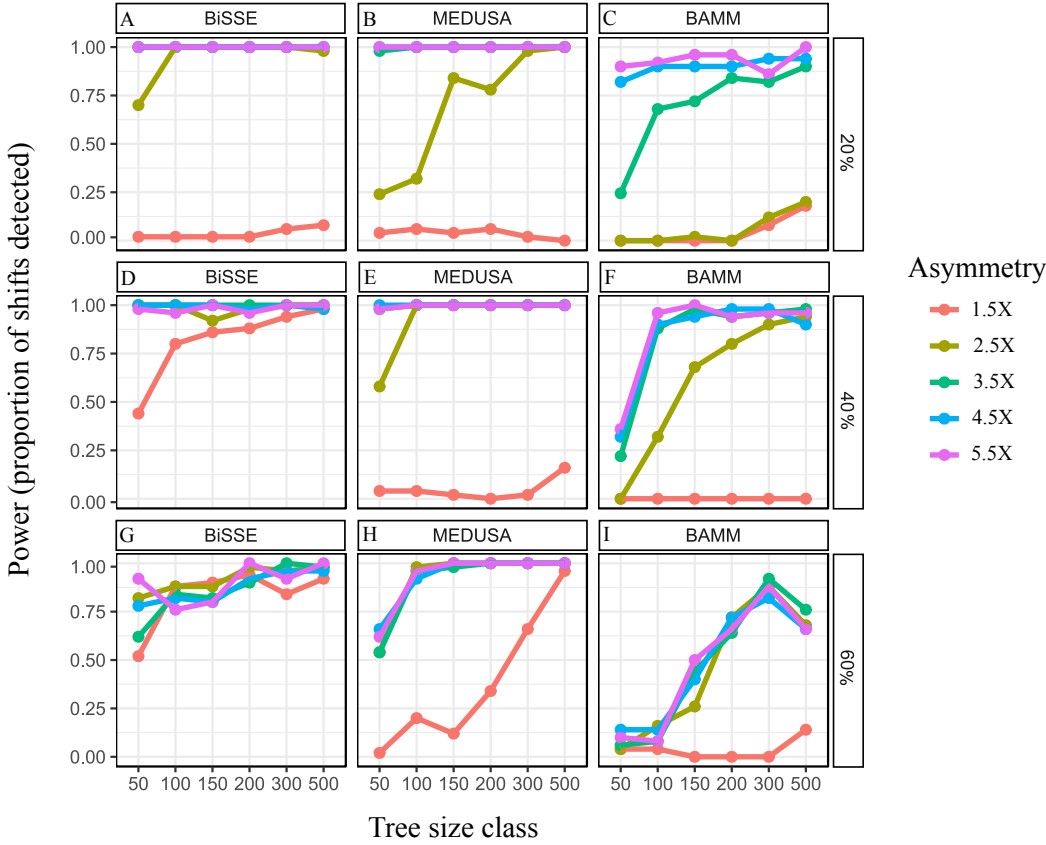

**Figure 2** **Effect of speciation rate asymmetry, relative *subtree* age and overall tree size.** Power, measured as the proportion of shifts detected, for BiSSE (A, D, G), MEDUSA (B, E, H) and BAMM (C, F, I), in trees from the Simulation Set 1. For BiSSE, the depicted results are when power was measured using 'approach 1' (see Materials and Methods, 'BiSSE'). All trees had the same *basetree* speciation rate of $\lambda_0$, but *subtree* speciation rate $\lambda_1$, $S_{age}$, and tree size differed. Trees were of the following size classes: $50 \pm 10$, $100 \pm 10$, $150 \pm 10$, $200 \pm 10$, $300 \pm 10$ and $500 \pm 10$. Speciation rate asymmetry varied from 1.5X to 5.5X. $S_{age}$ was 20% (Row 1), 40% (Row 2) or 60% (Row 3).

In the case of MEDUSA and BAMM, for $S_{ages}$ 20% and 40%, power was close to zero for an asymmetry of 1.5X irrespective of tree size, while power was very high (nearly 1) for all tree sizes when asymmetry was 4.5X or 5.5X (Figs. 2B, 2C, 2E, 2F). The exceptions were the set of BAMM analyses on trees of $S_{age}$ 40% (Fig. 2F), where 4.5X and 5.5X asymmetries had low power (less than 0.5) for very small tree sizes. For intermediate asymmetries (2.5X and 3.5X), power correlated strongly with tree size for both methods. At $S_{age}$ 60%, no asymmetry level resulted in uniformly high or low power across all tree sizes. Rather, there was a strong association between power and tree size for all asymmetries (Figs. 2H, 2I).

The number of *basetree* tips, *subtree* tips and the ratio of *basetree*:*subtree* tips (tip-ratio) for a given tree size class depended on $S_{age}$ (Supplemental Information 3). Both the number of *basetree* tips and tip-ratio bias decreased, but *subtree* size increased, with increasing $S_{age}$. The effect of these three parameters was explored further in Simulation 2.

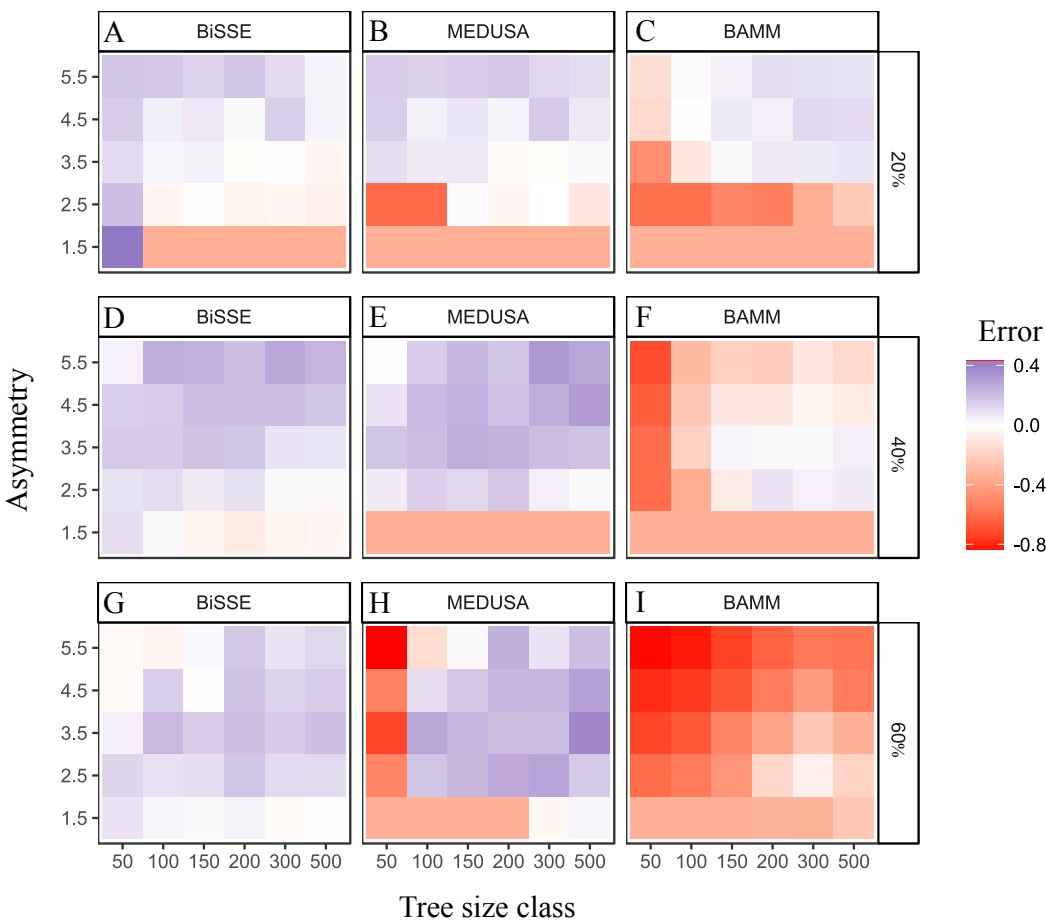

**Figure 3** **Error in estimation of speciation rate asymmetry.** Relationship between true and estimated speciation rate asymmetry for trees from Simulation Set 1: BiSSE (A, D, G), MEDUSA (B, E, H) and BAMM (C, F, I). Error = (Estimated asymmetry-True Asymmetry)/True Asymmetry. A value of 0 represents no error, values <0 represent underestimation and those >0 represent overestimation.

## Error in estimated speciation rate asymmetry in Simulation Set 1

Figure 3 depicts the relationship between true (i.e., simulated) and estimated asymmetry ratios. BAMM had a strong tendency to underestimate asymmetry (negative error values), whereas BiSSE tended to overestimate asymmetry (positive error values). MEDUSA generaly overestimated asymmetry, but tended to underestimate this when the true asymmetry was low and for lower tree sizes.

## Effect of tip-ratio bias and the number of tips in the *basetree* and *subtree* on power

Figure 4 depicts results from analyses of trees simulated in Set 2, all of which had a 2X asymmetry, overall size ranging from 50 to 800, and one of three tip-ratios, either biased towards the *basetree* (9:1) or the *subtree* (1:9 and 1:19).

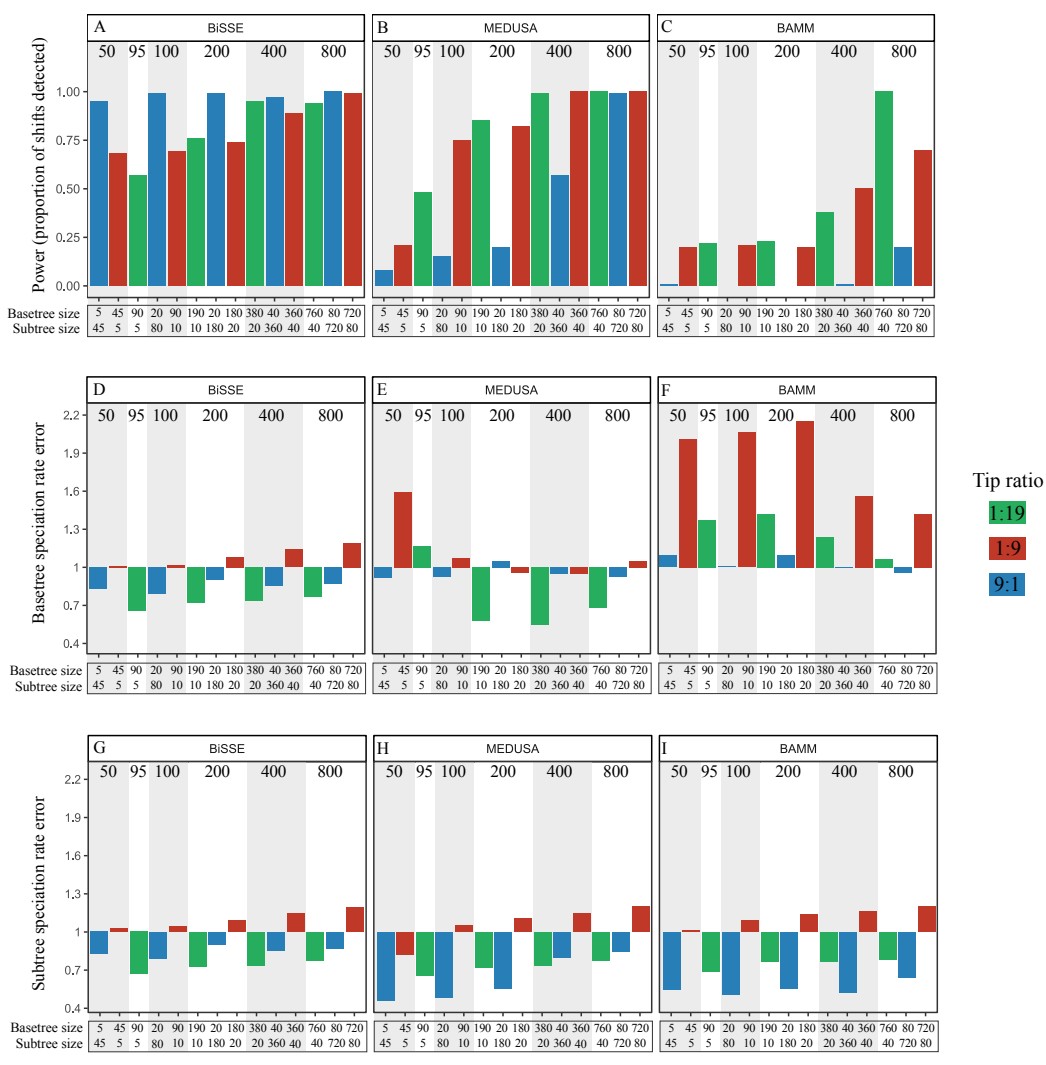

**Figure 4** **Effect of lineage size, tree size and tip-ratio.** Power, measured as the proportion of shifts detected, for BiSSE (A), MEDUSA (B) and BAMM (C), in trees from the Simulation Set 2. All trees had a 2X asymmetry, but belonged to one of five size classes ($50 \pm 10$, $100 \pm 10$, $200 \pm 10$, $400 \pm 10$ and $800 \pm 10$) and three tip-ratios, i.e., *basetree:subtree* tips. Bars are colour coded based on the tip-ratio. X axis labels indicate overall tree size and number of tips in the *basetree* (above) and *subtree* (below). Numbers above the bars indicate total tree size. The second and third rows depict error in estimation of *basetree* speciation rate $\lambda_0$ (D–F) and *subtree* speciation rate $\lambda_1$ (G–I). Error was the ratio of estimated/true $\lambda$. A value of 1 represents no error, values >1 indicate overestimation and values <1 indicate underestimation of $\lambda$. Note: We were unable to simulate trees with 1:9 ratio for $100 \pm 10$ size class, and therefore used a 1:8 ratio as an approximation.

### BiSSE

(Figure 4A): We present only the results from 'approach 1' of model selection because all three approaches produced similar results. Power was nearly 1 for all tree sizes with 9:1 ratio (blue bars), but ranged from ca. 0.7 to 1 among trees of 1:9 tip-ratio (red bars) and from ca. 0.60 to ca. 0.9 among trees with 1:19 tip ratio (green bars). For the latter two tip-ratios,

power tended to increase when *basetree* size (5, 10, 20, 40 or 80) increased, but did not increase when *basetree* size remained constant and only the *subtree* size increased (comparisons between adjacent green and red bars). Power was nearly 1 for all tree sizes when tip-ratio was *basetree* biased (9:1 tip-ratio, blue bars). Thus, power was generally higher for tip-ratio 9:1 than for 1:9 for the same overall tree size (comparison of adjacent blue and red bars).

### MEDUSA

(Figure 4B): Power was strongly correlated with overall tree size for all three tip-ratios. Power ranged from ca. 0.2 to 1 for the 1:9 tip-ratio (red bars), and from ca. 0.5 to 1 for the 1:19 tip-ratio (green bars). For smaller tree sizes, power increased when *basetree* size remained constant and *subtree* size increased, for e.g., comparison of trees with 5–45 and 5–90 *basetree-subtree* tips, or 20–180 and 20–380. Power ranged from ca. 0.1 to ca. 1 for the *basetree* biased tip-ratio (blue bars), and was overall higher for 1:9 compared to 9:1 (comparison of adjacent blue and red bars).

### BAMM

(Figure 4C): Power remained at ca. 0.25 for both *subtree* biased tip-ratios when tree size $\leq 200$ tips (red and green bars). For trees of the 400 size class, power was ca. 0.4 when there were 20 *subtree* tips, but increased to ca 0.5 when there were 40 *subtree* tips. Power was low ($>0.25$) for the *basetree* biased tip-ratio irrespective of tree size (blue bars). Power was highest for trees with 800 tips and *subtree* biased tip-ratios (0.75 or 1). Power was overall higher for tip-ratio 1:9 than for 9:1 (comparison of adjacent blue and red bars).

### Error in speciation rate estimates in simulation set 2

BiSSE had a tendency to underestimate $\lambda_0$ and $\lambda_1$ (error values <1) at ratios 1:19 (green bars) or 9:1 (blue bars), and overestimate these two parameters at 1:9 (red bars; Figs. 4D, 4G). MEDUSA had a tendency to underestimate $\lambda_0$ and $\lambda_1$ (error values <1) at 1:19 (green bars) and to overestimate $\lambda_0$ for the smallest trees at 1:9 ratio (red bars; Fig. 4E). At 1:9 ratio, $\lambda_0$ tended to be overestimated, while $\lambda_1$ tended to be underestimated. BAMM tended to overestimate $\lambda_0$ for all tree sizes and tip-ratios, and this effect was strongest at the 1:9 tip-ratio (red bars; Fig. 4F). All three methods tended to overestimate $\lambda_1$ when the tip-ratio was 1:9 (red bars), and underestimate this for the 9:1 (blue bars) and 1:19 (green bars) tip-ratios (Figs. 4G–4I).

## DISCUSSION

Previous studies have identified shortcomings specific to particular modelling approaches for estimation of rate shifts in phylogenies (e.g., *Rabosky, 2010*; *Davis, Midford & Maddison, 2013*; *Laurent, Robinson-Rechavi & Salamin, 2015*; *Rabosky & Goldberg, 2015*; *Gamisch, 2016*; *Moore et al., 2016*; *Kozak & Wiens, 2016*; *May & Moore, 2016*; *Wiens, 2017*; *Burin et al., 2018*; *Simpson et al., 2018*). We simulated large sets of trees where speciation and extinction rates remained constant throughout the tree (*basetree*), apart from an increase in speciation rate at a single node (*subtree*), and analyzed these trees using three widely used modelling approaches. We are therefore able to assess the relative performance of the

three methods, and identify issues that are common to these methods. We find that power to detect speciation rate shifts is strongly influenced by rate asymmetry and tip number for all methods.

## Effect of rate asymmetry

Not surprisingly, power increased as the speciation rate asymmetry increased, which has also been reported in other studies (e.g., *Davis, Midford & Maddison, 2013*; *Laurent, Robinson-Rechavi & Salamin, 2015*). All methods performed poorly when the *subtree* speciation rate increased by only 50% (1.5X) relative to the *basetree*, suggesting that moderate rate increases are difficult to detect. However, even with a high asymmetry of 5.5X, a significant proportion of rate shifts were undetected by all methods (Type II error), especially in smaller trees (Fig. 2). *Davis, Midford & Maddison (2013)* and *Gamisch (2016)* simulated complex evolutionary scenarios with multiple increases and decreases in diversification parameters at random points across the tree, and assessed the effect of overall tree size. They showed that BiSSE analyses on small trees are prone to high Type II error. *Laurent, Robinson-Rechavi & Salamin (2015)* tested the performance of MEDUSA for both simple scenarios with a single speciation rate shift and more complex scenarios with multiple shifts and mass extinctions. Interestingly, they found that overall tree size increased power in the complex scenarios (their Figure 5a), but not for the single rate shift scenario (their Fig. 3B). In their single rate shift scenario, power increased with the size of the lineage in which the rate shift occurred (i.e., the *subtree*). This suggests that power may be affected not by the overall tip number, but by the number of tips in the *basetree* or the *subtree*, both of which are correlated with overall tip number. We discuss this in more detail in the section '*Effects of lineage tip number and tip-ratio bias*'.

The performance of all three methods was generally similar at high asymmetry values, although BAMM tended to have lower power at $S_{age}$ 60% even with high asymmetry. Overall, BiSSE appears to perform better than other two –power tended to be high except when the lowest asymmetry was combined with the lowest $S_{age}$. MEDUSA appears to perform better than BAMM when asymmetry is low. When $S_{age}$ was 20%, BAMM rarely detected a rate shift at 2.5X asymmetry even in the largest trees. At 40% $S_{age}$, BAMM had lower power than the other two methods at 2.5X asymmetry, irrespective of tree size. Furthermore, BAMM performed worse than the other two when the second set of simulations with 2X asymmetry were analyzed (Fig. 4). Power was negligible for both BAMM and MEDUSA at the weakest asymmetry (1.5X) in Simulation Set 1.

Power should increase if the speciation rate asymmetry is more accurately estimated. This was apparent for BAMM and MEDUSA, which both strongly underestimated asymmetry for smaller trees and at lower asymmetry values (Fig. 3). BiSSE, on the other hand, tended to overestimate asymmetry (Fig. 3), but there was no clear relationship between error and power for this method.

## Effect of *subtree* age

The power of all methods was lower at $S_{age}$ 60% compared to $S_{ages}$ 40% and 20%. This is not due to tree size, because the simulated tree sizes were the same for all $S_{ages}$. However,

for a given asymmetry value and overall tree size, *subtree* sizes are necessarily larger for higher S$_{ages}$ (Supplemental Information 3), and therefore *basetree* sizes need to be smaller to accommodate the larger *subtrees*. Thus, *subtree* size, *basetree* size and tip-ratio bias were all influenced by S$_{age}$, and one or all of these factors may explain why power was compromised at S$_{age}$ 60% compared to the younger S$_{ages}$.

## Effects of lineage tip number and tip-ratio bias

The second set of simulations explicitly attempted to tease apart the effects of overall tree size, *subtree* size, *basetree* size and tip-ratio. We simulated three tip ratios, two of which were *subtree* biased (1:9 and 1:19) and the third, *basetree* biased (9:1). We varied *basetree* and *subtree* size from 5-720, and the overall tree size from 50-800. We were thus able to not only test whether tip-ratio bias affected power, but also compare power for trees with the same *subtree* or *basetree* size, but varying overall tree size. As expected, overall tree size generally correlated positively with power for all methods. However, although both 1:9 and 9:1 tip ratios had the same tip-ratio bias, these ratios differed in power within the same tree size class; this was the case for all methods (Figs. 4A–4C; comparison of red and blue bars). For BiSSE, there was either no difference in power between these two tip-ratios or power tended to be greater at 9:1, but power was always greater at 1:9 for MEDUSA and BAMM. Furthermore, power varied significantly among tree sizes for any given tip-ratio. This was so even when tree sizes were very large, and therefore when power may be expected to be uniformly high. This indicates that the number of tips in the lineages with distinct speciation rates (i.e., *basetree* and *subtree*) may play a stronger role than tip-ratio bias *per se*. Indeed, both *subtree* and *basetree* sizes independently had a strong effect on power for all three methods.

However, the methods differed in terms of how influential *subtree* and *basetree* size were. For BiSSE, power did not increase when tree size doubled at the same *basetree* size, for e.g., when *basetree* was 10 and *subtree* size changed to 190 from 90. If lineages on a tree differ in speciation rates, estimates of these rates should be more error prone in the case of smaller trees, because these lineages will be smaller. Generally, if lineages differ in diversification rates, the size of such lineages with distinct diversification parameters will be larger in larger trees. This may explain why power increases with tree size, as has been found here and in other studies (e.g., *Davis, Midford & Maddison, 2013*; *Laurent, Robinson-Rechavi & Salamin, 2015*; *Gamisch, 2016*). The effect of lineage size also explains results in *Davis, Midford & Maddison (2013)*, where power initially increased with increasing asymmetry but later decreased with further asymmetry increase, and this pattern was consistent for all overall tree sizes (their Fig. 1A). They concluded that the positive effect of asymmetry was counteracted by the negative effect of increasing tip-ratio bias as asymmetry increased. However, at higher asymmetries (and thus at stronger tip-ratios bias), the number of tips available for parameter estimation is likely to have been the limiting factor. For e.g., a tree with 500 tips (the largest tree size they simulated) and an asymmetry of 10X had a tip-ratio of 90:1, and thus one of 'character states' (i.e., all lineages sharing the same diversification rate parameter) would have been represented by <six tips. Therefore, we suggest that their results are better explained by the effect of the number of tips in each state rather than

tip-ratio *per se* (although we do not argue that tip-ratio has no effect), and that stronger asymmetries will *always* improve power of detection by BISSE as long as there are enough tips of every character state.

For MEDUSA, there was a tendency for increase in power when *subtree* size increased without a change in *basetree* size (Fig. 4B; comparison of red and green bars), indicating that overall tree size may partially offset the deleterious effect of small lineage size. This effect was also seen in the BAMM analyses, where the *basetree* remained at 40, but *subtree* size doubled.

Inferences can be drawn about the relative importance of *subtree* and *basetree* sizes by comparing 9:1 and 1:9 tip-ratios (Fig. 4; comparison of red and blue bars for the same tree size). For BiSSE, power tended to be higher for 9:1 (blue bars), suggesting that small *basetree* size is more detrimental compared to small *subtree* size. On the other hand, power was higher at 1:9 (red bars) than at 9:1 (blue bars) for both MEDUSA and BAMM, indicating that the three methods are differently affected by *basetree* and *subtree* sizes. Both BAMM and MEDUSA performed poorly at 9:1—while BAMM rarely detected rate shifts at this tip-ratio even in the largest trees, MEDUSA had very low power when trees had less than 400 tips. The low power maybe related to the fact that both methods strongly underestimated *subtree* speciation rates (Figs. 4H, 4I).

Given that BiSSE had the best power and BAMM the worst, it was not surprising that BiSSE had the lowest error and BAMM the highest. One may perhaps expect that speciation rates are more accurately estimated in the monophyletic *subtree* compared to the paraphyletic *basetree*. However, we found no strong differences in error between *subtree* and *basetree*. An exception was BAMM at 1:9, where error was clearly much higher for the basetree (comparison of red bars in Figs. 4F and 4I).

## Recommendations

In practice, a researcher intending to analyze diversification rate shifts may only have information about overall tree size, and not the sizes of lineages with distinct rates. In such cases, we agree with recommendations of Davis and colleagues (*Davis, Midford & Maddison, 2013*) who suggested that inferences based on BiSSE analyses of trees with fewer than 300 tips should be made very cautiously. We extend the same recommendation to BAMM and MEDUSA. Thus, if no rate shift is detected when using BiSSE, BAMM or MEDUSA on small phylogenies, users should be careful when concluding that speciation rate has been constant.

Davis and colleagues (*Davis, Midford & Maddison, 2013*) recommended cautious interpretation when analyzing datasets where <10% of species are of one character state. We conclude that the number of tips with a particular character state are a better predictor of power, rather than proportion, especially when using BiSSE. For instance, if 5% of the tips have a character state (i.e., 1:20 tip-ratio when there are only two character states), power of detection may not be compromised as long as there is strong rate asymmetry, there are at least 100 tips with the character state and the character state is not phylogenetically overdispersed. Generally, we suggest that tip-ratio may not be a big problem when analyzing very large trees (>1,000 tips).

## ACKNOWLEDGEMENTS

The authors thank Ranjit Kumar Sahoo for initial discussions. Comments from the editor as well as the very extensive and constructive critique by Michael R. May greatly improved the quality of the study.

### Funding

This work was supported by IISER Thiruvananthapuram intra-mural funding (through MHRD) and the INSPIRE Faculty Award to Ullasa Kodandaramaiah (DST/INSPIRE/04/2013/000476). The funders had no role in study design, data collection and analysis, decision to publish, or preparation of the manuscript.

### Grant Disclosures

The following grant information was disclosed by the authors:
IISER Thiruvananthapuram intra-mural funding.
INSPIRE Faculty Award to Ullasa Kodandaramaiah: DST/INSPIRE/04/2013/000476.

### Competing Interests

The authors declare there are no competing interests.

### Author Contributions

- Ullasa Kodandaramaiah conceived and designed the experiments, prepared figures and/or tables, authored or reviewed drafts of the paper, approved the final draft.
- Gopal Murali conceived and designed the experiments, performed the experiments, analyzed the data, contributed reagents/materials/analysis tools, prepared figures and/or tables, authored or reviewed drafts of the paper, approved the final draft.

### Data Availability

Kodandaramaiah, Ullasa; Murali, Gopal (2018): What affects power to estimate speciation rate shifts?. figshare. Fileset. https://doi.org/10.6084/m9.figshare.5331550.v1.

### Supplemental Information

Supplemental information for this article can be found online at http://dx.doi.org/10.7717/peerj.5495#supplemental-information.

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
