# Peer review of "What affects power to estimate speciation rate shifts?"

_PeerJ, doi:10.7717/peerj.5495_

## Round 0.1 · original submission · Major Revisions

Firstly, apologies for the time taken, it took us some time to find reviewers. In the end, we only found one. However, since this review seems informed and is very useful, I have decided to press on based on this review and my own reading.

The key point raised by the reviewer is that simulations mean little unless they are tied better to the assumptions of what they are exploring, in this case certain analytical approaches for studying speciation rates. My feeling is that this is an important point but perhaps reflects shortcomings in the way the paper is organised. To me, too much of the introduction focuses on how various programs are able to make inferences about how traits influence the rate of speciation when in fact the simulations and analysis presented in the paper seem to focus entirely on the inference of speciation rate.

To make the paper both more cohesive and publishable, I suggest reorganisation. The introduction needs to be written from the standpoint that it is useful to know how speciation rate changes and, underpinning this, is an ability to infer speciation rates from phylogenies. A brief section is then needed to explain why the inference of changes in speciation rate is apparently so difficult. You then need to explain that several programs exist that aim to do this as part of more extensive analyses of trait evolution. The reader needs to see what assumptions each package makes, its basic approach and what input format the data are given (for example, is the input a tree with node error bars, sequence data from which the tree is constructed or a fixed ‘known’ tree without errors, or something else?).

The above introduction sets the scene for an exploration of the power of each algorithm to detect changes in rate of speciation (sudden or gradual?), and aspects of the speciation process that impact this power. As far as I can see, this would negate the main issue seen by the Referee because the key question is simple: how is variation in branching rate best detected. The Referee also makes a lot of useful suggestions, most of which are sensible and therefore should be addressed.

·

Basic reporting

This manuscript is generally well organized and logically structured, the figures are high quality and very clear, and the manuscript generally fulfills basic reporting. I have only a few very minor concerns here:

1. The supplementary data archive is very thorough, and seems sufficient to recreate all of the major results. However, the code for simulating trees is relatively undocumented and difficult to interpret. You should provide more detail about the logic and details of this simulation in the code and manuscript (I will return to this point in the Experimental Design section). Additionally, a README file in the FigShare archive would help readers understand the workflow of the simulation and analysis, and would facilitate transparency and replication.

2. On line 29, you seem to attribute the "advent of phylogenetics" solely to Hennig 1965 (presumably you mean "phylogenetic inference", as the concept of phylogeny itself has much deeper historical roots). Certainly phylogenetic inference has a complicated origin, but if you are going to cite specific works then it would be reasonable to cite Edwards + Cavalli-Sforza 1963, Sokal + Sneath 1963, Camin + Sokal 1965, among others, as early examples of phylogenetic inference per se.

3. I disagree with your citation and interpretation of Hartmann, Wong, and Stadler (2010) on line 125. This misinterpretation underlies a more serious conceptual issue with the experimental design, so I will go into more detail on this concern in that section.

4. There are some grammatical and typographical errors; I am providing a details list below of the errors I identified:

- line 19: "For BiSSE, we assigned different character states for the lineages with different simulated speciation rates." This seems like a technical detail that does not need to be in the abstract.

- line 21: "... but had high Type II errors" -> "... but had high Type II error rates".

- line 22: "algorithms" -> "methods", just for consistency.

- line 22: "identified" should probably be "identify", to maintain the present tense when referring to the results

- line 23: "both" -> "all three"

- line 24: "performed" -> "perform", as per comment about line 22

- line 50: "models estimate" -> "methods estimate". "Models" allow us to compute probabilities, but do not themselves estimate parameters. "[statistical] methods" use probability calculations provided by models to estimate parameters.

- line 51: The use of "information" and the description of the types of "information" here are confusing. Clade age and branch lengths are not very distinct from each other. I would just say along the lines of "These estimates are based on divergence times(/branch lengths) and the topological distribution of species among clades", or something similar.

- line 54: What does "diversity" mean here? All lineages are elements of diversity. Are you referring to diversity-dependent models? Please clarify.

- line 73: "without a priori information on character states". It would be more accurate to say that the character-independent diversification model are agnostic about the mechanism of rate variation, not that they are agnostic about the character state. The source of rate variation need not be a character in the standard phylogenetic sense, but could be, for example, a unique historical event. Invasion of an island (an event) may lead to increased diversification rates; this is distinct from the idea that living on islands (a character state) increases diversification rates.

- line 80: "estimations" -> "estimates", plus comments about BAMM, above.

- line 86: "were" -> "are"

- line 92: "the of estimates" -> "the estimates"

- line 93: "the BiSSE" -> "BiSSE" or "the BiSSE method"

- line 96: "estimation" -> "estimates"

- line 124: "simulation of branching process conditioned on number" -> "simulation of branching processes conditioned on the number"

- line 145: ";;" -> ";" within the parentheses.

- line 150: "Bage" -> "B_age"

- line 164: "for a sample trees" -> "for a sample of trees"? Also, why just a sample of trees? MCMC is not especially costly: you should do this for each analysis.

- line 176: I assume "pure birth-death process" -> "birth-death process"?

- line 231: "(Type II error, red curves in Figure 1)" Should this refer to Figure 2?

- line 284: "readers" -> "researchers"?

Experimental design

The goal of this research is to understand the statistical behavior of some of the most popular methods in the study of phylogenetic/macroevolutionary lineage diversification. This is very well motivated: the lineage diversification field is currently in a bit of disarray, and a thorough presentation of how these methods behave would be extremely timely and valuable to the community! Unfortunately, the execution of the simulation study has serious flaws, which I detail below.

In order to receive my recommendation for publication, you should develop a reasonable strategy for simulating data that is as close to the generating models as possible, so that the impact of model misspecification on power and parameter estimates is minimal. The BAMM and MEDUSA analyses should also be performed to a higher standard (see the specific sections below regarding my concerns).

1. The data are simulated in a very ad-hoc way.

You note that "simulation of branching process [sic] conditioned on number of tips can lead to bias in the diversification rate (Hartmann, Wong, and Stadler 2010)". This is a misinterpretation of the Hartmann paper: the issue they identify is with an incorrect procedure for forward simulating trees conditional on the number of species under a birth-death process (it is incorrect to terminate the simulation as soon as there are exactly #n species in the tree). Their point is that, when you simulate data incorrectly, parameter estimates based on those simulations will be biased. They show that it is straightforward to 1) correctly simulate trees conditional on the number of species, and 2) generate unbiased parameter estimates on the resulting trees. This might seem irrelevant, but the issue that Hartmann, Wong and Stadler identify is _directly_ applicable to your simulations: ad hoc simulation procedures result in biased parameter estimates.

One appealing property of model-based inference methods is that, in theory, as the amount of data increases to infinity, parameter estimates converge to the true value (for well-behaved, so-called "consistent" statistical methods). However, this is only true if the inference model and the generating model are the same: obviously, if the inference model is wrong, then parameter estimates can never be perfect. For example, if you simulate trees conditional on the number of species, and estimate parameters conditional on the age of the tree, parameter estimates will be compromised. This consideration implies that ANY discrepancy between the generating and inference models can compromise statistical performance. This is concerning because your simulation procedure is not a "generating model" for any of the inference models that you use. It is therefore very difficult to understand whether the issues you ultimately identify are properties of the methods (statistical behavior) or are just an artifact of your simulation procedure.

In short, I do not think the experimental design you have chosen is justifiable, and it obscures the interpretations of your results. In general, if you want to understand the statistical performance of a method, you need to simulate data under a model that resembles the inference model as much as possible. Otherwise, if the method performs poorly, we do not know whether it's because of a property of the method or because of the simulation procedure! This is effectively an uncontrolled experiment. One solution to this problem would be to simulate data under the models assumed by each method. I know this is an impossibly tall order: MEDUSA in fact does not describe such a model, and the model assumed by BAMM has become increasingly ambiguous and may be impossible to simulate under.

Another solution would be to focus on "empirical performance" rather than "statistical performance". An alternative approach to simulation studies is to try to understand how biologically realistic violations of model assumptions compromise parameter estimates. In this case, we may simulate data under some well-motivated but non-generating model, and check parameter estimates under the inference model to see how accurate they are. This will give us some idea about how reliable the parameter estimates are given some biologically realistic part of parameter space. This, however, requires some sense of "biologically realism", which is difficult to quantify, and is quite different from the approach of the current manuscript!

(As a technical asside, I will note that TESS does not use forward simulation (line 121), nor does TreeSim use backward simulation when conditioning on age (line 133). These details aren't really critical to the present work anyway, so I think they can be safely omitted.)

2. Your exploration of parameter space further obscures behavior.

Specifically, because you simulate conditional on a small number of tree ages (25 and 15), tree size (number of taxa) is conflated with rate shift size. The size of the tree is proportional to the amount of information available to estimate parameters, so is the observation that power increases as rate shift size increases an artifact of tree size?

It would be better to adjust the tree age for a particular combination of simulation parameters so that the expected number of species hits some target value; you may iterate over target values to achieve differences in tree size. So that tree size and rate shift size are not correlated, it is critical that the expected number of species is for the entire simulation procedure, not just the "basetree". This can be achieved by Monte Carlo simulation: for a given tree age and set of simulation parameters, simulate many trees and compute their average age; repeat until you find an age that provides the target number of species. In this way you could simulate small trees with large rate shifts by decreasing the age of the tree.

As far as the combinations of parameters you choose to simulate, e.g., the ratio of basetree speciation rate and subtree speciation rate: how did you choose these values and why? For example, you use only a single---relatively low---value of the extinction rate; we know these methods are sensitive to the (relative) extinction rate (May and Moore 2015), so I consider this a serious limitation. Your lambda_1/lambda_0 ratios are quite finely divided: this isn't a bad thing by itself, but is probably very inefficient. If you want to explore more extinction rates, then using ratios at 0.5 intervals should be sufficient. I would also like to see ratios greater than 2.5. Especially if you are demonstrating that the methods have low power for small trees, it would be nice to know how large the rate shifts have to be to get good power for these trees!

3. The analysis of your simulated data is inadequate.

One major concern here is that you are using a very loose definition of a "true positive"; in particular, on line 150, you note that you compute "the proportion of trees in which a significant rate shift was detected." Is simply detecting a rate shift sufficient for MEDUSA and BAMM? Especially for methods we know can have high false positive rates, it seems like we should be checking whether or not the inferred location of the rate shift is correct. A "true positive" should be required to identify not just the number but the location of rate shifts!

3.1 BAMM (lines 160-172).

In general there needs to be more detail provided on these BAMM analyses. What priors did you use and why? It looks as if you used "default" exponential(1) priors on the speciation and extinction rates, and on the expected number of events. (These details need to be in the main text, as some of your readership may not be able to determine this from your supplementary files.) Is the behavior of BAMM sensitive to these priors? If so, then your current results would be uninformative. If you really want to understand how BAMM behaves in general, you need to give some thought to prior sensitivity (as you would with any Bayesian method). The developers of BAMM do not advocate using these default values, and provide a function (setBAMMpriors in the BAMMtools package) for generating not-unreasonable priors.

What does it mean for BAMM to "report >0 rate shifts" (line 171)? I can see from your supplementary material that you used the number of shifts with the highest posterior probability (the maximum a posteriori estimator, or MAP). Why? This is an especially bad choice, since a) the MAP estimate masks prior sensitivity and posterior uncertainty, and b) your prior expected number of rate shifts is 1. What happens if you use the posterior mean number of rate shifts? Again, it seems important to determine whether the _location_ of the rate shift was correctly identified!

It seems from your supplementary files that your BAMM analyses were only performed for 25 out of 100 simulated trees per combination of parameter settings. Is this a mistake in the supplementary data? If not, this needs to be explicit in the manuscript (though it would be preferable to do all 100 analyses).

3.2 MEDUSA (lines 173-178).

On line 175, you say: "We specified the model of tree evolution to be a Yule process as the model did not converge for mixed or pure birth-death process". This is quite troubling. First, how pervasive was this phenomenon? Did you try different initial conditions (the init argument)? I tried MEDUSA on a few (100) of your trees and it seemed to work fine! If the method doesn't work for computational reasons, it is a bit unfair to constrain the method in some way if you are really interested in the statistical behavior. I would recommend getting the method to work; if you cannot do so, then do not present these results!

Validity of the findings

Given the concerns outlined above, it is difficult to make much of the research findings. The interpretations of the results, taken at face value, are adequate. However, the advice given for methods developers and users of these methods is unclear, ambiguous, and/or unsafe---major revisions to the advice provided in the discussion section are required before I would recommend acceptance of this manuscript.

1. Results

For the reasons described above regarding the simulation design, it is difficult to make much of the results. All four variables (speciation rate symmetry, relative subtree age, overall tree age, and number of tips) are conflated because of the design of the simulation experiment, so we're effectively unable to understand factors that impact power. The simulated data depart to an unknown degree from each of the inference models, so it is difficult to interpret the biases in parameter estimates. Accommodating my advice about the experiment design, above, would alleviate both of these issues.

Also, from Figure 5 it looks like you are only reporting BAMM results for lambda_1/lambda_0 >= 1.5, whereas you are reporting results for this ratio of 1.1 for the other BiSSE. Is this intentional?

2. The advice given to method developers and users is unsound, or otherwise requires major clarification.

You say starting line 271: "the effects of tip number (both subtree and overall tree), and associated parameters such as relative subtree age, may be the most serious and universal issues for modelling approaches, and development of future methods should focus attention on rectifying these." It's not clear to me what you mean, here. Tip number and relative subtree age are properties of the data, not the method. I don't agree that these are issues of modelling approaches; rather, I see them as limitations on methods imposed by the data! Could you clarify how you expect method developers to use your results?

On lines 279-281, you say: "Users should be extremely cautious when using BiSSE, BAMM or MEDUSA on small phylogenies and should avoid using small, incomplete phylogenies to test hypotheses of rate shifts." Given that your results seem to suggest these methods have low power for small trees, I disagree with this sentiment; rather, I think users should be mindful about how they interpret the outcomes of hypothesis tests. Failing to reject a null hypothesis is not the same thing as confirming the null hypothesis! In this sense there is no "danger" of using a small tree: failing to reject rate constancy does not mean that there is no rate variation! Likewise, parameter estimates for small trees should have wide confidence/credible intervals and be interpreted accordingly. Could you clarify why you think we should be especially cautious? Excess caution does not seem to be supported by your results.

On line 281, you say: "When testing hypotheses of change in speciation rates, we recommend that users explore results from multiple datasets, such that the relative age of the clade of interest with respect to the entire tree varies." Unless I am misunderstanding your intentions, I find this advise dubious and impractical (many clades will not have multiple "datasets", depending on your definition of a dataset). Could you provide more detail on what you are suggesting? If you think that the power of the method depends strongly on the relative age of the subtree and are suggesting that users try datasets with different relative ages, this advise seems dangerously close to "fishing for significance".

Starting line 283, you say: "In analyses of trait-dependent diversification rates, we also encourage readers to report results from traditional sister-group species richness comparisons such as the richness Yule test (Paradis, 2012), and recently developed non-parametric tests such as FiSSE (Rabosky & Goldberg, 2017)." Again, I find this advice of questionable statistical merit. What is the purpose for doing this? One interpretation is that you are suggesting a simple sensitivity analysis: since all these methods differ slightly in the assumptions they make, then if all the methods agree, the result is "robust" to different assumptions---in principle, this would be a fine recommendation. The danger of this approach is an inflated sense of confidence. Certainly the results of multiple methods applied to the same dataset will be correlated with each other? I imagine that for your own simulations, when BiSEE correctly identified a rate shift, that both MEDUSA and BAMM were more likely to also identify that rate shift? The resulting "pseudo-consilience" runs the risk of providing the user with a false sense of the actual support for the hypothesis. If you are going to give this advice, you need to clarify what you are suggesting and give some thought to how multiple statistical analyses of the same data should be interpreted.

Additional comments

no comment

---

## Round 0.2 · Minor Revisions

This paper is now much improved. However, the very thorough review from Reviewer 1 has made a lot of constructive suggestions that will improve the paper further. Although these are classed as 'minor revisions' there is a reasonable amount to do and I would urge the authors to pay close attention to what is being asked and to adhere as closely as they can to the suggestions.

·

Basic reporting

I have a few concerns about reporting:

1. Figures 3 and 4 are hard to interpret.

The goal of figure 3 (in my interpretation) is to show that these methods generate biased estimates of the relative rate of speciation between the subtree/basetree. However, the fact that the "target" ratio changes for each row makes it difficult to see patterns in the bias as a function of the relevant parameters. Rather than simply showing the median estimate of the rates in each cell, it may be more clear to show the mean (or median) percentage error in each cell (the average difference between the estimate and the true value, divided by the true value). The advantage of this measure is that it has the same "target" value, 0%, regardless of the ratio. Also, I think the term "expected speciation asymmetry" is not quite correct: this is really the true value, and not the expected value of an estimator or a random variable (which is what "expected" immediately makes me think of).

Figure 4 is very difficult for me to wrap my head around. I do not think that barplots are the right way to display this information. I would use Fig. 1 from Davis, Midford, and Maddison (2013), which shows power as a function of the tip-ratio, as a template here.

2. Miscellaneous

There are also some typographical errors and other minor issues related to reporting. Here is a (non-exhaustive) list of minor issues:

Abstract: the authors mention that lineage-diversification methods allow "researchers to correlate diversification rate shifts with causal ecological factors", but in the introduction they describe the various hypotheses that these methods are intended to address (traits, key innovations, biogeography, climate, etc.). To describe the latter set as "ecological factors" is a bit too narrow: I would recommend ammending the abstract to reflect that these methods are useful for more than just testing ecological hypotheses.

line 41: "'key innovations')(e.g." -> these two parentheticals should be combined.

line 44: e.g.Jansson -> e.g. Jansson (missing space)

line 58: "For long" is a fairly unnatural way to begin a paragraph. I might replace this with "Historically," or some equivalent conjunction.

line 67: "likelihood model" should just be "model", or "stochastic model" or "birth--death--transition model". Stochastic models provide ways of computing probability functions that can be used as likelihood functions.

line 68: "( Maddison" -> "(Maddison" (extra space)

lines 69-72: The language used to describe likelihood-based inference and model comparison is a bit murky, and should be clarified. The likelihood score is proportional to the probability of the data given a set of parameters. Maximum likelihood methods find the values of the parameters that maximize the likelihood score. The values of the parameters that maximize the likelihood are the maximum-likelihood estimates of the parameters. The maximum likelihood score can be used to compare competing models using, for example, the likelihood ratio test or AIC. (To be more specific: "likelihood scores" are not "estimated simultaneously for" parameters.)

line 96: "Markov Chain Monte Carlo" -> "Markov chain Monte Carlo" (chain is not capitalized)

line 106: "BiSSE has been shown to be affected..." This phrase is a bit vague. I would say that "The ability of the BiSSE method to detect state-specific diversification rates has been shown to depend on tree size and tip-ratio bias: the method seems to behave well with large trees and low tip-ratio biases" (or perhaps something more concise).

line 160: "detecte" -> "detect"

line 172: "mu_1 )" -> "mu_1)" (extra space)

line 223: "Sage" -> "age" should be subscripted.

line 348-352: "lineage" is used in an unusual way. Typically, "lineage" refers to a branch in a birth--death model, but here it is being used to describe the branches associated with a particular character state. These sentences should refer to the number of lineages in each state.

line 375: "When testing hypotheses of character-dependent diversification, users typically have information through ancestral state reconstructions on the size of lineages that potentially have distinct diversification rates." I think this sentence is not quite correct. When rates of diversification depend on the state, "standard" discrete character models (on which ancestral states depend) will fail: this observation is actually what motivated the development of the BiSSE family of models (see Maddison's 2006 brief communication in Evolution).

line 383: I don't know what the authors mean when they say "the character state is not polyphyletic." This statement should be clarified.

Experimental design

I have a few concerns about the description of the simulation procedure. In general, the description needs to be more explicit. For example, the authors state that (line 144):

"As this method produces trees with varying number of tips, we simulated 1000 trees each time and used the function tess.nTaxa.expected from the same package to select the best tree out of the 1000 trees. This was done by comparing the expected number of tips for every 1 unit time with the simulated tree using a correlation test. The tree with the maximum correlation coefficient was chosen as the best tree."

It is unclear what "each time" and "every 1 unit time" means. It's also not clear what the goal of the correlation test is, and how it is actually used: is it simply to pick the simulated tree that has the number of tips that is closest to the expected value?

Later, the authors describe varying the age of the tree (B_age) to obtain the targeted tree size (line 154). The authors should describe the motivation and execution of this procedure in more detail.

Because this study is primarily about the behavior of methods under simulation, I think it's critical that the description of the simulations makes it clear what the authors did, and why they did it. This section should be written with enough detail that a reader could recreate these simulations without having to look at the author's code. I also think that sections (1) and (2) can be combined into a single section, since section (1) exists solely to elucidate details of section (2)---the current ordering is a bit counter-intuitive.

2. Power

In Methods section (3), the authors describe computing the power as "the proportion of trees in which a significant rate shift was detected at the node where the subtree was attached to the basetree". However, it's not clear how this relates to BiSSE (which does not locate rate shifts, but rather tests for equality between state-specific diversification rates), or BAMM (which in principle could be used to test a hypothesis about a rate shift occurring on a given branch, but the authors do not describe what they actually do).

I would recommend that the authors explicitly state their definition of power for each of the three methods: for BiSSE, this will be a likelihood ratio test for equal or different speciation rates; for MEDUSA, this will be whether a multi-rate model is preferred and locates the rate shift on the correct branch; for BAMM, this will be some test that should be detailed (what the authors chose is unclear).

3. BiSSE analyses

Methods section (3a) describes comparing the unrestricted BiSSE model (lambda_1 != lambda_0, mu_1 != mu_0, q_01 != q_10) to a model where the speciation rates are restricted to be the same (lambda_1 == lambda_0). I am not sure that this is the right comparison: in the simulation, the extinction rates are fixed to be equal, so it is hard to imagine that the restricted model is the "true" model. It's not clear from the simulation what the "true" transition parameters should be equal or different, since the simulated data do not have rate shifts that are generated by a Markov process.

In any case, I believe the authors should explore some additional BiSSE models. In particular, I think the most relevant model comparisons are between (lambda_1 != lambda_0, mu_1 == mu_0, q_01 != q_10) and (lambda_1 == lambda_0, mu_1 == mu_0, q_01 != q_10). Because the "true" transition model is unclear, the authors might also compare (lambda_1 != lambda_0, mu_1 == mu_0, q_01 == q_10) and (lambda_1 == lambda_0, mu_1 == mu_0, q_01 == q_10). I would be surprised if these comparisons made much of a difference, but it feels odd to base the performance on two "false" models that assume the extinction rates for each state are unequal (mu_1 != mu_0).

4. Miscellaneous

Again, I find the use of "expected" (line 203) to be a bit confusing. The authors should simply call this the "true" asymmetry. For the asymmetry experiments, I also think that the authors should be computing relative error (as they do for their tip-ratio experiments) or percentage error for the asymmetry, which will make systematic patterns in the bias more obvious.

Validity of the findings

I have no real concerns about the authors' interpretations of their results.

Additional comments

This manuscript has improved substantially from the initial version. The research is well-motivated and timely, and will help a broad range of consumers of these popular methods interpret their results. However, I have a few (minor) outstanding concerns.

I would like to see the authors provide more explicit details about the simulation design and improve the clarity of their figures. There are a few remaining concerns with the data analysis section, relating to the definition of power and the choice of BiSSE models to compare. I provide more details for my concerns in the basic reporting and validity sections of my review.

---

## Round 0.3 · accepted · Accept

I feel the authors have dealt well with the points raised and, all in all, this is now a very much improved manuscript.

#